# Effect of Essential Nutrients on Roots Growth and Lifespan of Huanglongbing Affected Citrus Trees

**DOI:** 10.3390/plants9040483

**Published:** 2020-04-09

**Authors:** Alisheikh A. Atta, Kelly T. Morgan, Said A. Hamido, Davie M. Kadyampakeni

**Affiliations:** 1Southwest Florida Research and Education Center, University of Florida, 2685 SR 29 N, Immokalee, FL 34142, USA; conserv@ufl.edu (K.T.M.); shamido@ufl.edu (S.A.H.); 2Citrus Research and Education Center, 700 Experiment Station Rd, Lake Alfred, FL 33850, USA; dkadyampakeni@ufl.edu

**Keywords:** *Candidatus Liberibacter asiaticus*, citrus greening, essential nutrients, root length density, root lifespan, soil pH

## Abstract

Understanding citrus tree root development and dynamics are critical in determining crop best nutrient management practices. The role of calcium (Ca) and magnesium (Mg), manganese (Mn), Zinc (Zn), and boron (B) on huanglongbing (HLB) affected citrus trees’ root growth and lifespan in Florida is not fully documented. Thus, the objective of this study was to determine the impact of foliar and ground-applied essential nutrients on seasonal fine root length density (FRLD; diameter (d) < 2 mm) and coarse roots (d > 2 mm), FRLD dynamics, root survival probability (lifespan), and root-zone soil pH of HLB-affected sweet orange trees. Results indicated that Ca treated trees budded on Cleopatra (Cleo) and Ca and Mg combined treatments on Swingle (Swc) rootstocks significantly increased seasonal FRLD of fine (< 2 mm) and coarse roots. The highest median root lifespan of Ca treated trees was 325 and 339 days for trees budded on Cleo and Swc rootstocks, respectively. In the second study, the coarse roots showed a significantly higher reaction to the nutrition applied than the fine roots. Meanwhile, the 2× (1× foliar and 1× ground-applied) treated trees showed a significantly higher median root lifespan compared to the other treatments. Thus, the current study unwraps future studies highlighting the combined soil and/or foliar application of the above nutrients to stimulate FRLD and improve root lifespan on HLB-affected sweet oranges with emphasis on root-zone soil pH.

## 1. Introduction

Huanglongbing (HLB, or citrus greening) is the most destructive endemic disease of Florida citrus production, with no identified cure insight [1,2]. The Asian citrus psyllid (ACP, *Diaphorina citri* Kuwayama) is a widespread pest throughout Florida is an important pest of citrus as it transmits the phloem-limited bacteria *Candidatus Liberibacter asiaticus* (*Ca*. Las) [3,4,5]. Once a citrus tree has been infected with Ca. Las, the tree becomes unprofitable within a few years [3,6]. According to previous studies, HLB inhibits root growth, reduces water and nutrient uptake, and increases leaf and fruit drop, resulting in deformed fruit with an unpleasant flavor [3,7,8]. Moreover, root damage caused by pathogen infestation could reduce the tree’s water and nutrient uptake, making it more susceptible to biotic and abiotic stresses [8,9,10]. Therefore, nutrient management or modification through fertilization to impact nutrient availability is an essential cultural practice to control plant disease and is a fundamental component of sustainable crop production [3,10,11]. 

The knowledge of FRLD distribution relative to the above-ground biomass for citrus trees is well documented [12]. Maximum FRLD in the pre-HLB era ranged from 0.53 cm.cm^–3^ for Swingle citrumelo to 2.02 cmcm^–3^ for trifoliate orange (*Poncirus trifoliata*). Although the citrus root system is estimated to account for more than 65% of above-ground dry mass [12,13] the root system of HLB-affected citrus trees is usually poorly developed and new root growth can also be inhibited [14]. Studies indicated that 30%–50% of roots of HLB-affected citrus trees are impaired before canopy symptoms appear and 70%–80% of root loss could be prevalent as citrus deprived of intensive cultural management to pacify abiotic and biotic stress [10,15]. The optimum distribution of the root system is mainly restricted by water and nutrients available in soil profiles [16,17]. However, it has been reported that HLB develops an imbalance of nutrient concentrations, which cause nutrient toxicity for some nutrients, including Cu or deficiency for others such as Ca, Mg, Mn, and Zn [18]. Therefore, nutrient supply is an imperative aspect of disease control because nutrients influence plant resistance, pathogen vigor, growth, and associated factors [19,20]. 

Ground-applied fertilizers are subject to unfavorable soil processes such as precipitation as forms that are unavailable to plants, leaching, and runoff [21,22]. Therefore, split ground and foliar-applied fertilizer improve nutrient use efficiency and reduce both nutrient leaching and application cost [23,24,25]. Moreover, HLB-affected trees are limited in soil nutrient uptake because of the associated decline in FRLD [12,16,26]. The role of adding essential nutrients in improving citrus growth in general and FRLD specifically over the seasons, along with root dynamics, and root lifespan, are not well understood. Hence, the following hypotheses were formulated: (1) foliar and or ground-applied essential nutrients improve fine root growth of HLB-affected citrus trees, as trees have nutritional requirements, which have to be met for healthy root growth and (2) adding essential nutrients also boost seasonal FRLD lifespan highlighting root-zone soil pH because nutrient deficiency reduces root growth and lifespan. The objectives of the following two studies were to evaluate whether ground in addition to foliar application of essential macronutrients (N, Ca, and Mg) or micronutrients (Mn, Zn, and B) can reverse the decline of the FRLD, monthly FRLD pattern, and root lifespan of HLB-affected sweet orange trees.

## 2. Results

### 2.1. Effect of Secondary Macronutrients on FRLD 

Roots of d < 1 mm showed the least FRLD as compared to the other fine root d > 1 mm and coarse roots. No significant variation on FRLD with d < 1 mm was observed among the secondary macronutrient treatments in any growing seasons of 2018 and 2019 (Table 1 and Table 2). Significantly greater FRLD d >1 mm was observed for trees budded on Cleo rootstocks that received Ca nutrition. Similarly, the current study indicated that the FRLD of citrus trees budded on either Cleo or Swc rootstocks were lower at the beginning of the study in spring 2018. There was a significant rootstock × secondary macronutrient interaction effect in the summer season of the same year for the FRLD of the fine root d > 1 mm. Since the interaction of the rootstock × secondary macronutrient was significant, data were analyzed separately by rootstock. The FRLD d > 1 mm contributed the highest proportion and was significantly higher during summer for Ca nutrient treated trees for trees budded on Swc than Cleo rootstocks. In summer 2018, about 11%, 34%, 6%, and 10% of FRLD d > 1 mm of trees budded on Cleo accounted for the untreated control trees, Ca, Mg, and Ca and Mg combined treated trees, respectively. Similarly, 11%, 21%, 18%, and 19% of the FRLD of trees budded on Swc ascribed to the untreated control trees, Ca, Mg, and Ca and Mg combined treated trees, respectively.

The Ca and Mg combined nutrition contributed significantly to fine and coarse root growth on trees budded on Swc rootstocks. It was only Ca nutrition on trees budded on Cleo rootstocks, which persisted to impact the FRLD in the fall season of 2018. Thus, the FRLD d > 1 mm was significantly greater for Ca nutrition treated on trees budded on Cleo rootstocks as compared to the other treatments. The Ca nutrition treatment stimulated significantly greater FRLD for trees budded on Cleo rootstocks and Ca and Mg combined nutrition contributed significantly to FRLD on trees budded on Swc rootstocks. 

### 2.2. Effect of Micronutrients on FRLD

The second study indicated that there was a significant variation for seasonal root growth at the beginning of the study with the lowest FRLD during the fall season of 2018. While treatment effect was observed in the summer season of the 2018 growing season with significantly higher FRLD for the soil and foliar-applied micronutrients only for coarse root FRLD (Table 3). The highest micronutrient rate 3× (1× foliar and 2× ground-applied) treated trees showed a reduction in the FRLD essentially in response to the decline of the root-zone soil pH generated as the result of the S, which encapsulates the ground-applied micronutrient formulation. The root-zone soil pH in the three years of the study indicated the highest soil acidity in soils of trees that received the highest S encapsulated micronutrient rate. Invariably, the impact of nutrition on FRLD was persistent during the next fall season. In summer 2018, about 48%, 29%, 23%, and 42% of the FRLD accounted for the 0× (untreated control), 1× (foliar only), 2× (1× foliar and 1× ground-applied), and 3× (1× foliar and 2× ground-applied) treated trees, respectively. A similar pattern was observed during the second year of 2019 growing season that in summer 2019, about 1.3%, 9%, 13%, and 7% of the FRLD accounted for the untreated control 0× (untreated control), 1× (foliar only), 2× (1× foliar and 1× ground-applied), and 3× (1× foliar and 2× ground-applied) treatments, respectively. During the second year, the decline in FRLD d < 1 mm was an indication of sensitivity to root-zone soil pH.

### 2.3. The FRLD Dynamics

The total FRLD of Hamlin citrus trees budded on Cleo and Swc rootstocks was significantly lower in spring than the summer and fall seasons of the first year, 2018, with the peak root growth observed in April/May and August/September months (Figure 1). Thus, the Ca nutrition generated significantly greater FRLD of the secondary macronutrients for trees budded on Cleo rootstocks and the Ca and Mg combined nutrition for trees budded on Swc rootstocks. This result, following the above result on the FRLD of fine roots with d >1 mm and coarse roots contributed the highest portion of the FRLD and was significantly higher during the summer and fall than the spring season. The FRLD dynamics and distribution resumed during the second year with a slight decrease in the fall cold season. Similarly, the FRLD of Hamlin citrus trees budded on Cleo was continuously influenced by Ca nutrition and on Swc rootstocks for trees treated with Ca and Mg combined nutrition.

The micronutrient experiment showed similar root growth patterns as the secondary macronutrient study that the fall FRLD had significantly lower than the summer and fall season in the first year, 2018 (Figure 2). The highest micronutrient rate, 3× (1× foliar and 2× ground-applied), showed significantly lower FRLD followed by foliar 1× (foliar only), 2× (1× foliar and 1× ground-applied), and 0× (untreated control) treated trees, respectively. The decline in root growth in the highest rate was attributed to the lower pH level created as the result of the high rate of ground-applied S encapsulated metallic Mn and Zn micronutrients. The 1× (foliar only) treatment promoted above-ground biomass but was not able to promote root growth; hence, the FRLD growth was lower than the untreated control. The 2× (1× foliar and 1× ground-applied) treatment had a significantly higher effect on FRLD; this could be because of foliar and ground-applied nutrition managed to balance the above and below-ground biomass of the trees. The untreated control trees with relatively low above-ground biomass had higher FRLD as compared to the highest rate. The foliar and ground-applied nutrition persisted to act on FRLD that the highest rate had significantly lower FRLD than the rest of the treatments. While the only 1× (foliar only), 2× (1× foliar and 1× ground-applied) micronutrient treatments showed a greater increase on FRLD during the early of 2019 growing seasons. In summer 2019, the 2× (1× foliar and 1× ground-applied) treatment showed a slight decrease in FRLD growth, which could be justified that the soil accumulation of the S caused to drop the root-zone soil pH. On the other hand, the bio-nutrient accumulation of the 1× (foliar only) applied treatment shown a significant increase in the FRLD during the late summer season of 2019.

### 2.4. Root Survival Analysis

The above results indicated that plant nutrition could promote FRLD of Hamlin trees budded on either Cleo or Swc rootstocks. The next step was to study how long the roots survived from emergence to death. The coarse roots of the untreated control trees budded on Cleo had about 3× shorter root lifespan than trees of the untreated Hamlin citrus trees budded on Swc rootstocks regardless of the treatments (Figure 3). Secondary macronutrients showed a significant effect on root survival (lifespan) according to the Gehan–Breslow statistic (Table 4 and Table 5). Thus, the reaction of the root lifespan to the secondary macronutrient was significantly higher for trees budded on Cleo than Swc rootstocks. The median root lifespan of the untreated control, full Ca dose, full Mg dose, and the combined half Ca and half Mg doses for trees budded Cleo rootstocks were about 61, 325, 189, and 285 days and for trees budded Swc rootstocks were about 245, 339, 248, and 220 days, respectively.

The second study indicated that S encapsulated micronutrient applications could significantly stimulate FRLD as well as the root lifespan with the emphasis in the root-zone soil pH anomalies. Like FRLD, the highest micronutrient rate increased the root lifespan of Valencia trees for probable the same reason, i.e., elevated soil acidity (Figure 4). The median root lifespan of the 0× (untreated control), 1× (foliar only), 2× (1× foliar and 1× ground-applied), and 3× (1× foliar and 2× ground-applied) treated trees were 254, 249, 316, and 191 days, respectively (Table 5 and Table 6). Thus, the highest micronutrient rate showed 49%, 75%, and 93% less median root lifespan as compared to the 0× (untreated control), 1× (foliar only), and 2× (1× foliar and 1× ground-applied) treatments. Meanwhile, the micronutrients had a significantly higher impact than the secondary macronutrients on the median root lifespan of Valencia than Hamlin citrus trees provided that both trees budded on Swing rootstocks (Table 5 and Table 6). However, the secondary macronutrients promoted more fine roots, while the micronutrients increased coarse roots on Hamlin and Valencia citrus trees, respectively.

### 2.5. Soil Acidity

The root-zone soil pH results on Hamlin citrus trees indicated a significant three-way interaction effect of nitrogen × rootstocks × secondary macronutrients except during the first season of the study, 2018 (Table 7). Hence, the results of the soil pH in Table 7 were displayed separately by rootstocks at the three soil depths. The soil pH on the untreated control trees was persistently and significantly higher than the treated trees. The soil pH on Ca treated trees had no significant variation as compared to the untreated control trees in any season of the three years because no S was applied. The soil pH was significantly higher in the topsoil layer (0–15 cm) than the two lower soil depths (15–30 and 30–45 cm) during the entire study.

The results of soil pH on Valencia citrus trees indicated a significant interaction effect of soil depths × micronutrient interactions during the entire study (Table 8). During the spring season of 2017, the soil pH varied across the three soil depths in response to the S contained in S encapsulated Mn and Zn encapsulated micronutrient rates. Except for the highest micronutrient rate 3× (1× foliar and 2× ground-applied), the root-zone soil acidity was significantly higher for the rest of the treatments. This pattern was similar in the three soil depths. Yet, the soil pH was significantly lower at the topsoil layer as compared to the two lower soil depths pertained to the S accompanied with the micronutrients. The soil pH showed a similar pattern as the previous season, except that no significant variation was observed for all micronutrient treated as compared to the untreated control trees at the lowest soil depth (30–45 cm).

In the second year, there was a significant interaction effect soil depths × micronutrients on the root-zone soil pH. In this year during both seasons, the soil pH was significantly lower in the soil treated trees, 2× (1× foliar and 1× ground-applied) and 3× (1× foliar and 2× ground-applied) than the 1× (foliar only) and the 0× (untreated control) treated trees. The two upper soil layers (0–15 and 15–30 cm) had the highest soil acidity for trees that received the highest ground-applied micronutrient rate in both seasons of the year 2018. This indicated that there was leaching of S to the bottom of the soil that caused the drop in the root-zone soil pH.

The soil acidity increased with time and decreased with the increase in the soil depth. A similar trend was observed in the spring of the third year, 2019 as the previous seasons with ground-applied fertilizer treated trees showing the lowest soil pH as compared to the only foliar and untreated control trees. The increase in the soil acidity was mainly attributed to the increase in the S obtained from the encapsulated metallic Mn and Zn treatments. Therefore, ground-applied dolomitic limestone (Oldcastle, Atlanta, GA, USA) at a rate of 1.2 kg tree^−1^ was applied in Mar. 2019 to lower the acidity of the soil. Consequently, the soil pH showed an increment of about 8%, 13%, and 30% of soil pH as compared to the same season of the previous year in the 1× (foliar only), 2× (1× foliar and 1× ground-applied), and 3× (1× foliar and 2× ground-applied) micronutrient treated trees, respectively.

## 3. Discussions

The primary reaction to the Ca. Las associate problems of HLB-affected citrus trees is the decline of the FRLD [7,15,28]. A very low seasonal FRLD of roots with d < 1 mm was observed during spring and fall season of both years on trees that had received secondary macronutrients. Similar FRLD root classes and dynamics had been reported on young citrus ‘Valencia’ sweet orange trees on Swingle citrumelo rootstock [1,7] and 5-year-old ‘Hamlin’ and ‘Valencia’ orange trees in reaction to a different irrigation method, soil depth, and distance of the root from the tree [10]. The FRLD of root d > 1 mm had relatively greater density as compared to the rest of the other fine roots. Like previous studies, our study indicated that HLB significantly impacted the roots of sweet orange trees. However, rootstocks and the combination of essential nutrients showed a significant effect in reversing the deleterious effect of HLB that caused the citrus FRLD deterioration. Ca nutrition for trees budded on Cleo and Ca and Mg combined nutrient treated trees budded on Swc rootstocks significantly improved the FRLD of fine roots d > 1 mm. The variation in FRLD density was associated with the growth flush growth stage [7,17,29], moisture [16,17,30], and fertilization [17,29].

On the other hand, the micronutrient on Valencia citrus trees had a significant effect on the FRLD d < 2 mm. The results indicated on FRLD had a positive impact on the trees that received lower than the highest S encapsulated micronutrient rates. The highest S encapsulated micronutrient rate was the key factor in deteriorating the FRLD d < 1 mm during the late study seasons. The detrimental impact was probably caused by the elemental S, which upon dissociating with water created acidic soil in the root-zone environment. Consequently, the decline in FRLD had, in turn, a detrimental effect on the above-ground biomass (i.e., canopy volume and leaf area index, data not shown). Thus, the low pH-induced availability of soil cations was the justification for improved nutrient uptake fertilization management in the HLB-affected citrus industry. The monthly root growth dynamics displayed low FRLD at the beginning of the spring season and increased through April/May and decreased in May/June and again increased in late summer July/August. This pattern was also repeated similarly in the second year, portraying a bimodal FRLD growth pattern. The double peaks of root growth were the manifestation of the completion of the above-ground biomass of the trees and the beginning of more resource allocation to the root system. Previous studies indicated that the peak of root growth patterns was partly attributed to the phenology of the trees [27,30] in response to soil moisture, nutrient acquisitions, and other growth-promoting factors [7,17,27].The current study indicated that the trees Ca and Ca and Mg combined nutrients had a positive impact on the FRLD dynamics; while Mn and Zn had also a significant effect on FRLD with emphasis on root-zone soil pH management. The decline in FRLD and yield was partly attributed to HLB-induced damage of fine roots, which led to vulnerability to other biotic and abiotic stresses [1,12,15].

The trees budded on Swingle had a longer lifespan (3× more root lifespan) than trees budded on Cleo rootstocks irrespective of the secondary macronutrient treatments. Yet, trees budded on Cleo rootstocks were more reactive to Ca nutrition and Ca and Mg combined nutrition on Swc rootstocks on their FRLD. Meanwhile, Ca improved root median lifespan 5.0× and 1.4× as compared to their corresponding untreated control trees on Cleo and Swc rootstock, respectively. The median root lifespan for trees treated with full Ca dose, full Mg dose, and the combined half Ca and half Mg doses were about 5.3×, 3.1×, and 4.7× and 1.4×, 1.1×, and 0.92× for trees budded on Cleo and Swc rootstocks, respectively. This indicated that Ca nutrition had a significant impact on the median root lifespan regardless of the type of rootstocks. Therefore, Ca nutrition not only promotes FRLD of Hamlin citrus trees but also enabled roots to have a longer root lifespan in response to the secondary macronutrients treatments. Similarly, higher rates of secondary macronutrients promoted higher root survival probability. In the second study, the micronutrients significantly increased the Valencia citrus root lifespan when the trees received 2× (1× foliar and 1× ground-applied) treatments. However, the accumulation of soil S affected root lifespan over time in response to the increase in the root-zone soil acidity. Thus the median root lifespan was 1.2×, 1.3×, and 0.7× for the trees that received 1× (foliar only), 2× (1× foliar and 1× ground-applied), and 3× (1× foliar and 2× ground-applied) treated trees as compared to 0× (untreated control), respectively. The median root lifespan of healthy Valencia citrus trees was reported to be ≈ 113 days [8] and ≈ 95–105 days for fine roots d < 0.5 mm and 226–259 days for fine roots > 1 mm [1,27,31]. 

The root-zone soil pH in the Hamlin citrus root-zone was maintained in the lower to upper sixes during the entire of the secondary macronutrient study. This results in a favorable root growth environment on Hamlin citrus trees. Recent field studies indicated that keeping soil pH in the 5.5–6.5 resulted in greater root density and nutrient uptake [10]. On the other hand, the reduced lifespan of the highest micronutrient rate could be because of the drop in root-zone soil pH. The drop in the root-zone soil pH was caused by the dissociation of S in the soil water that came along the S encapsulated Mn and Zn nutrients. Under the acidic root-zone (pH < 5.0), soil minerals such as Al^3+^ solubilize and are released into the soil solution to the extent that affects root growth [32,33,34]. Meanwhile, extended exposure of high H^+^ activity in the root solution can disrupt the electrochemical gradient by increasing plasma membrane permeability and increasing Al^3+^ toxicity thereby restricting fine root growth with an influence on plant metabolism, plant nutrient uptake, movement, and utilization [32,33]. The fine root growth < 2 mm was relatively higher with micronutrients than and the lifespan of these roots was better with secondary macronutrient treatments. These results should prompt further study of combined secondary macronutrients and micronutrients on FRLD and root lifespan of HLB-affected sweet oranges highlighting the root-zone soil acidity.

## 4. Materials and Method

### 4.1. Description of the Study Site 

The studies were carried out starting in 2017 at the University of Florida, Southwest Florida Research and Education Center (SWFREC) near Immokalee, FL (26.42° N and 81.43° W). The sweet oranges (cv. Hamlin and Valencia) were planted in April 2006 on Immokalee fine sand, which is a poorly drained soil in the Flatwoods, containing sandy marine sediments with slopes < 2 percent and classified as sandy, siliceous, hyperthermic Arenic Haplaquods with a Spodic horizon lying within 1 m from the ground surface [16,35]. The trees were planted on a double-row of 13.5 m-wide raised bed, 6 m and 3m spacing between rows and trees, respectively (Figure 5B).

### 4.2. Experimental Design and Treatments

Two experiments were conducted between 2017 and 2019. The first study included ‘Hamlin’ sweet orange trees (*Citrus sinensis* (L.) Osbeck) budded on Cleopatra (*Citrus reshni*) or Swingle citrumelo (*Citrus paradisi* Macf. and *Poncirus trifoliata* (L.) Raf.) rootstocks. The second study included ‘Valencia’ sweet orange (*Citrus sinensis* (L.) Osbeck) trees budded on Swingle (Swc) rootstocks. Laboratory results of leaf samples taken prior to the start of the experiment performed by quantitative real-time polymerase chain reaction (qPCR) indicated that both sets of trees were positive for the HLB associated bacterium, Ca. Las with Ct value of 25.3 ± 0.25 and 25.9 ± 0.19 of the Hamlin and Valencia trees, respectively. Therefore, the trees in the current study were 100% infected with Ca. Las The first study was conducted as a split-plot design comprised of two rootstocks, two nitrogen rates (168 and 268 kg ha^−1^), untreated control, and three secondary macronutrients (Ca or Mg at 45 kg ha^−1^ and Ca and Mg combined at 22 kg ha^−1^ each) replicated four times. The second study had a similar design as the former except that the Valencia trees budded on Swc rootstocks received four treatments with a base (1×) rate of micronutrients (Mn (9 kg ha^−1^), Zn (9 kg ha^−1^), and B (2.2 kg ha^−1^)) instead of the secondary macronutrients as recommended by the University of Florida, Institute of Food and Agricultural Sciences for citrus nutrition. Therefore, treatment 1 had sub-sub plots of untreated control trees. Treatment 2 had only foliar-applied micronutrient treatments at a 1× rate. Trees receiving treatment 3 had micronutrients applied to the soil and leaves at 1× rate each. Trees receiving treatment 4 received a foliar application at 1× rate micronutrients and 2× rates to the soil. The ground-applied encapsulated Mn and Zn contained S to reduce the root-zone soil pH thereby enhancing the availability and uptake of these micronutrients by the tree roots. Thus, the treatments yielded applications of 0× (untreated control), 1× (foliar only), 2× (1× foliar and 1× ground-applied), and 3× (1× foliar and 2× ground-applied) as previously recommended rates [36,37].

The N was fertigated as a split biweekly application from February to November resulting in 20 applications per year. Meanwhile, all trees received 168 kg ha^−1^ of K_2_O as a split fertigation application along with the N. The secondary macronutrients and micronutrients treatments were applied three times yearly coinciding with early spring (March), summer (June), and late summer (September) leaf flush seasons [38,39]. The treatments were applied using a truck-mounted sprayer motor pump (Hypro corporation, New Brighton, MN) [37]. The volume of the solution applied to each of the sub-sub plots was determined and calibrated by a stopwatch. Aluminum Spray GunJet|AA(B) 43L-AL4 (Spraying Systems Co., Glendale Heights, IL) with a liquid maximum pressure capacity of 5515.81 kpa (kilopascal) was used to uniformly distribute the solution across the sub-plots.

The trees were irrigated daily with irrigation durations determined using smart irrigation apps for androids (http://smartirrigationapps.org/). SmartIrrigation apps are tools established to determine irrigation schedules for particular crops (i.e., avocado, citrus, cotton, peanut, strawberry, and vegetables) using reference evapotranspiration from Food and Agricultural Organization (FAO) Penman–Monteith and crop coefficient based on the real-time meteorological data obtained from the Florida Automated Weather Network at the experimental site [40,41,42]. The microsprinkler irrigation was accommodated with a 172.4 kpa irrigation pump at 45 L h^−1^ with Max-14 (Maxijet Inc., Dundee, FL, USA) fill-in blue deflector of 360° × 14 emitter per tree [22,37,43].

### 4.3. Minirhizotron Tube Installation

Eight transparent acrylic minirhizotron tubes were installed per treatment in March 2017 at 0.5 m either to the east or west of the tree trunk, perpendicular to the tree row (Figure 5A). Holes of relatively equal size to the tubes were augured using a 7.5 cm internal diameter (i.d.) auger to reduce soil disturbance and root damage. The tubes were inserted at a 45° angle to the ground and to a perpendicular depth of 45 cm [29,44,45] where most of the citrus trees’ roots are concentrated [12,26,46]. The bottom of the tubes was sealed with a plastic cup, the top end was covered with a detachable cap to impede the entry of water into the tubes. The tubes extending above the ground were covered with black plastic tape to avoid exposure of the roots to light that would affect root growth.

### 4.4. Data Collection

Nondestructive minirhizotron root data were collected and processed using a root scanner system (CI-600 Root Growth Monitoring System, Fa. CID, Camas, WA, USA) at the beginning of the study and every month for two years. The roots visible around the minirhizotron (21 × 19 ≈ 400 cm^2^) tubes were traced to estimate their respective length using RootSnap Version 1.3.2.25 Release software (2011–2013 CID Bio-Science, Inc. Camas, WA, USA). The total monthly root length was divided into the area covered by the image of the camera to which the roots were visible to determine the FRLD. The monthly FRLD data were grouped based on traditional root classification: fine roots (d < 2 mm) and coarse root (d > 2 mm). The fine roots were also divided into three sub-class (<0.5 mm, 0.5–1, and 1–2) [30,47]. The yearly average root data were regrouped into three seasons: spring (Jan.–May), summer (Jun.–Sep.), and fall (Oct.–Dec.) for analysis.

Soil samples were collected from three locations at three soil depths: 0–15 cm, 15–20 cm, and 30–45 cm. The samples were collected within ≈ 0.5–1 m radius from the tree trunk canopy using 2.5 cm i.d. soil core sampler in March and August of each year. The samples were collected from three different locations under a tree canopy and pooled into a sampling bag according to their depths. A total of twenty-four soil samples per treatment of three depths were collected in each plot from the four blocks. The root-zone soil pH was determined by adding 1: 2 soil to water ratio by volume using an AR 15 pH meter (Accumet^®^ Research, Fisher Scientific, Lakewood, CO, USA).

### 4.5. Statistical Analysis

The FRLD data were analyzed using repeated-measures analysis PROC GLM Mixed Model procedures SAS 9.4 (SAS Institute, Cary, NC, USA). Data subjected to spatial and temporal variability were tested for the four underlying statistical model assumptions: linearity, independence, normality, and homogeneity of variance. For F-tests with a statistical difference at (*p* ≤ 0.05) was used and the Tukey–Kramer honestly significant difference (HSD) grouping range test was used to compare the means. Twelve roots per tree of fine roots > 1 mm were randomly selected and traced from the time of emergence until their decline for root survival analysis on data collected during 2018 and 2019 growing seasons. The mean, 25^th^, median (i.e., 50^th^), and 75^th^ percentiles of root lifespan was determined using the sigma plot 14 (SigmaPlot 14, Systat Software, San Jose, CA, USA). The root survival analysis was performed with the Kaplan–Meier survival model to test for significant differences between survival curves.

## 5. Conclusions

Several studies have reported that HLB significantly reduced FRLD before any other symptoms appear on citrus trees. Essential nutrition type, method of application, rate, and frequency of applications had been devised in addition to the rootstock-scion association to reverse the impact of the HLB-induced problem on root growth and the lifespan of citrus trees. The current study showed that split foliar and ground application of the macronutrient (N and K) in addition to the secondary-macro and micronutrients had a significant impact on FRLD of coarse roots and root d > 1 mm as compared to roots d < 1 mm and their respective lifespan. Even though micronutrients showed a promising impact on the fine roots, the S obtained from the S encapsulated metallic Mna and Zn micronutrients reduced the initiation and lifespan of fine roots with d < 1 mm. Therefore, the study demonstrated that further research is needed on the combination of secondary macronutrients and micronutrients for HLB-affected sweet orange trees with emphasis on the root zone soil pH to stimulate if the fine root with d < 1 mm growth and their lifespan of sweet oranges can be improved.

## Figures and Tables

**Figure 1 plants-09-00483-f001:**
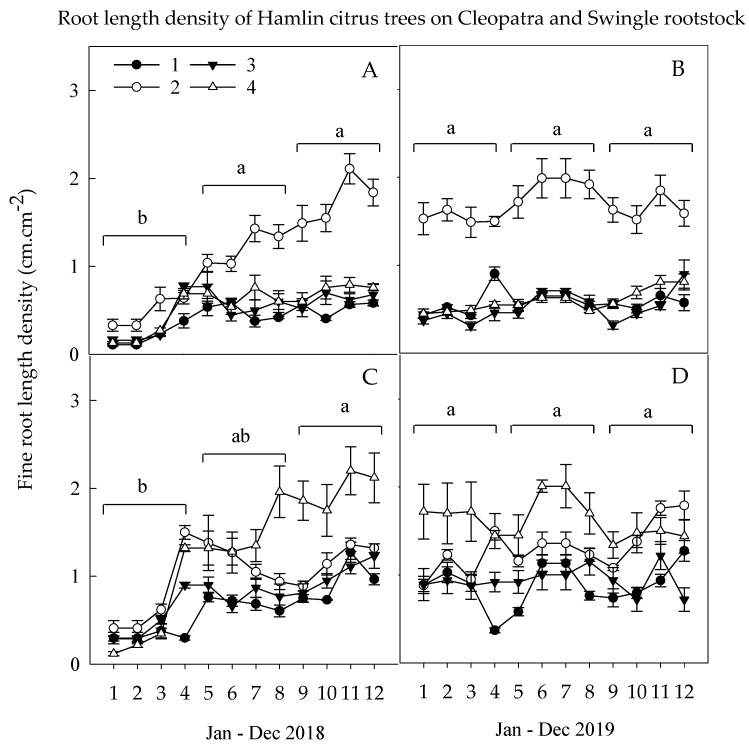
Root length density dynamics of citrus tree cv. Hamlin budded on Cleo (**A**,**B**) or Swc rootstocks (**C**,**D**) during spring (Jan.–May), summer (Jun.–Sep.), and fall (Oct.–Dec.) of 2018 and 2019 growing seasons. Treatments: untreated Control (1), full Ca dose (2), full Mg dose (3), and half Ca and half Mg doses (4), (full dose = 45 kg ha^-1^). The average seasonal FRLD are the mean values of (*n* = 8 trees) ± standard error of the mean (SEM).

**Figure 2 plants-09-00483-f002:**
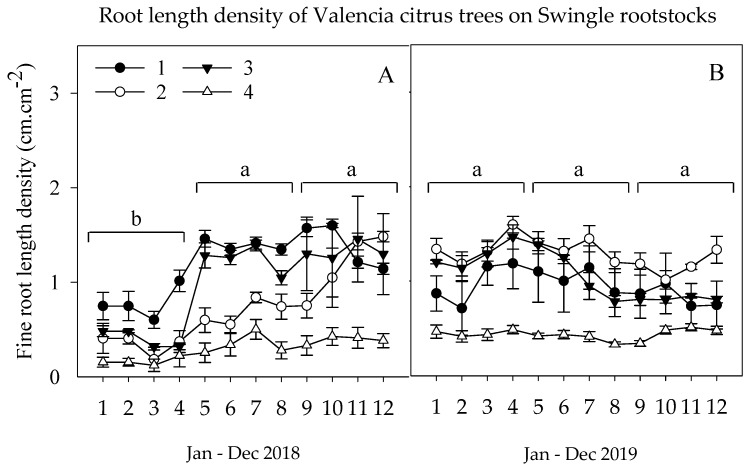
Root length density dynamics of citrus (Cv. Valencia) on Swc rootstocks during spring (Jan–May), summer (Jun.–Sep.), and fall (Oct.–Dec.) of 2018 (**A**) and 2019 (**B**) growing seasons. Treatments: (1) 0× (untreated control), (2) 1× (foliar only), (3) 2× (1× foliar and 1× ground-applied), and (4) 3× (1× foliar and 2× ground-applied), (1× = 9 kg ha^-1^ of metallic S encapsulated Mn and Zn each and 2.2 kg ha^-1^ of B). The average seasonal FRLD means values of (*n* = 8 trees) ± SEM.

**Figure 3 plants-09-00483-f003:**
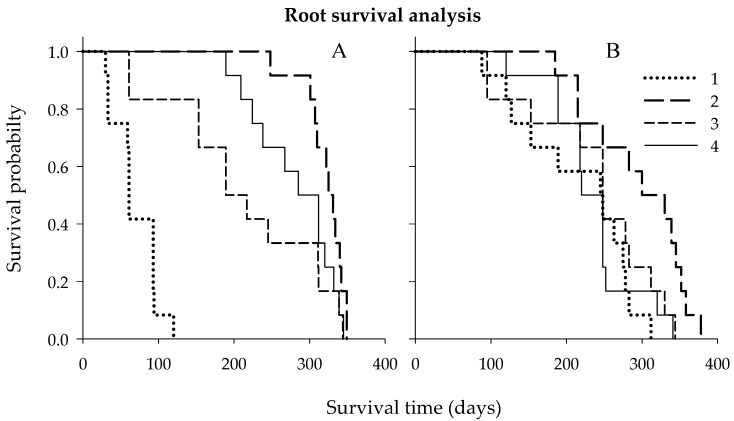
Kaplan–Meier root survival (lifespan) curve for citrus trees cv. Hamlin budded on Cleo (**A**) or Swc (**B**) rootstocks during the 2018 and 2019 growing seasons. Treatments: untreated control (1), full Ca dose (2), full Mg dose (3), and half Ca and half Mg doses (4), (full dose = 45 kg ha^−1^).

**Figure 4 plants-09-00483-f004:**
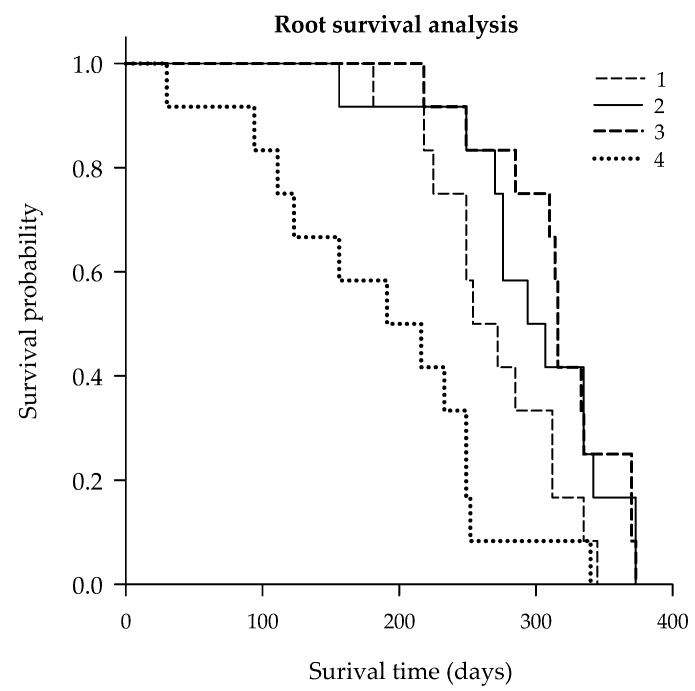
Kaplan–Meier survival curve for FRLD of citrus trees cv. Valencia budded on Swc rootstocks over 24 months follow up during 2018 and 2019 growing seasons. Treatments: (1) 0× (untreated control), (2) 1× (foliar only), (3) 2× (1× foliar and 1× ground-applied), and (4) 3× (1× foliar and 2× ground-applied), (1× = 9 kg ha^−1^ of metallic S encapsulated Mn and Zn each and 2.2 kg ha^−1^ of B).

**Figure 5 plants-09-00483-f005:**
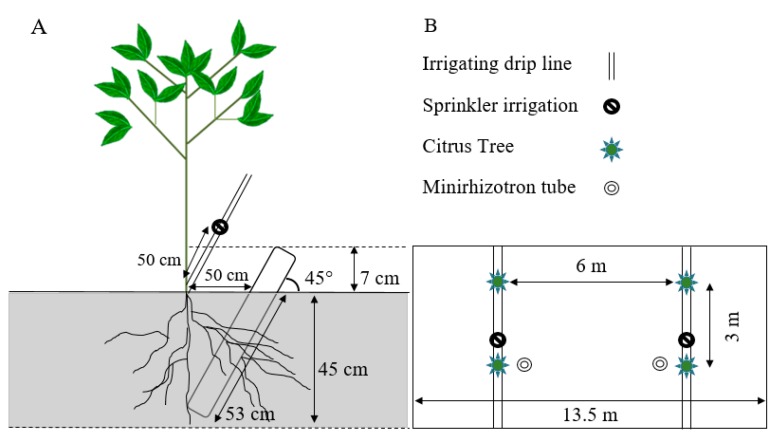
Schematic presentation of the citrus tree, minirhizotron tube, and irrigation sprinkler set up side view (**A**) and top view (**B**) of 13.5 m-wide bed, 6 m and 3 m spacing between rows and trees, respectively. The minirhizotron was located on the raised two-row beds and the sprinklers along the tree line at a right angle to each other from the tree and each 50 cm away from the trunk of the tree.

**Table 1 plants-09-00483-t001:** Analysis of variance (ANOVA) of the effect of rootstock, nitrogen, and ground-applied secondary macronutrients (Ca and Mg) on root length density of HLB-affected Hamlin citrus trees during 2018–2019 seasons.

	2018				2019			
	Root Diameter (mm)							
Factorial effect ^1^	<0.5	0.5–1	1–2	>2	<0.5	0.5–1	1–2	>2
T	_NS_ ^2^	_NS_	**	***	_NS_	_NS_	_NS_	***
R	_NS_	_NS_	_NS_	_NS_	_NS_	_NS_	_NS_	***
N	_NS_	_NS_	_NS_	_NS_	_NS_	_NS_	_NS_	_NS_
S	_NS_	_NS_	***	**	_NS_	_NS_	*	***
T × R	_NS_	_NS_	_NS_	_NS_	_NS_	_NS_	_NS_	_NS_
S × R	_NS_	_NS_	***	*	_NS_	_NS_	**	**
N × R	_NS_	_NS_	_NS_	_NS_	_NS_	_NS_	_NS_	_NS_
N × S	_NS_	_NS_	_NS_	_NS_	_NS_	_NS_	_NS_	_NS_
T × N	_NS_	_NS_	_NS_	_NS_	_NS_	_NS_	_NS_	_NS_
T × S	_NS_	_NS_	_NS_	_NS_	_NS_	_NS_	_NS_	_NS_
T × R × N	_NS_	_NS_	_NS_	_NS_	_NS_	_NS_	_NS_	_NS_
N × R × S	_NS_	_NS_	_NS_	_NS_	_NS_	_NS_	_NS_	_NS_
T × N × S	_NS_	_NS_	_NS_	_NS_	_NS_	_NS_	_NS_	_NS_
T × S × R	_NS_	_NS_	_NS_	_NS_	_NS_	_NS_	_NS_	_NS_
T × N × S × R	_NS_	_NS_	_NS_	_NS_	_NS_	_NS_	_NS_	_NS_

^1^ Factorial effects: T = season of the year, R = Rootstocks, N = nitrogen rate, and S = Secondary macronutrients, reading was taken every month from Jan. to Dec. of 2018 and 2019. ^2^ NS, *, **, and *** represent non-significant or significant at *p* ≤ 0.05, < 0.01, and < 0.0001, respectively.

**Table 2 plants-09-00483-t002:** The effect of rootstock, nitrogen and ground-applied secondary macronutrients (Ca and Mg) on root length density (FRLD) of HLB-affected Hamlin citrus trees during 2018–2019 seasons.

Root Diameter (mm)	Spring 2018							
	<0.5		0.5–1		1–2		>2	
Rootstocks ^1^	Cleo	Swc	Cleo	Swc	Cleo	Swc	Cleo	Swc
Treatments ^2^	Root length density (cm·cm^−2^)							
1	0.011 ^3^	0.046	0.034	0.011	0.061	0.060	0.089	0.250
2	0.033	0.012	0.011	0.034	0.057	0.118	0.375	0.404
3	0.035	0.022	0.041	0.038	0.062	0.132	0.141	0.242
4	0.015	0.012	0.004	0.030	0.005	0.078	0.140	0.333
Significance ^4^	_NS_	_NS_	_NS_	_NS_	_NS_	_NS_	_NS_	_NS_
Treatments	Summer 2018							
1	0.001	0.025	0.016 ab	0.034 a	0.043 b	0.058 a	0.330	0.436 b
2	0.060	0.020	0.155 a	0.011 a	0.357 a	0.064 a	0.488	0.734 ab
3	0.020	0.031	0.002 b	0.044 a	0.024 b	0.086 a	0.369	0.480 b
4	0.026	0.033	0.023 ab	0.046 a	0.048 b	0.135 a	0.403	1.077 a
Significance	_NS_	_NS_	*	_NS_	***	**	_NS_	***
Treatments	Fall 2018							
1	0.015	0.076	0.037	0.054	0.049 b	0.183	0.315	0.522
2	0.040	0.028	0.073	0.023	0.391 a	0.136	0.817	0.830
3	- ^5^	0.004	0.025	0.010	0.085 b	0.120	0.523	0.741
4	-	0.019	0.025	0.075	0.065 b	0.185	0.519	1.307
Significance	_NS_	_NS_	_NS_	_NS_	***	_NS_	_NS_	_NS_
	Spring 2019							
1	0.009	0.044	0.017	0.027	0.125 b	0.191	0.216	0.420 ab
2	0.018	0.030	0.058	0.020	0.518 a	0.175	0.581	0.616 ab
3	0.005	0.063	0.004	0.043	0.112 b	0.233	0.268	0.373 b
4	-	0.001	0.015	0.009	0.133 b	0.298	0.259	0.974 a
Significance	_NS_	_NS_	_NS_	_NS_	**	_NS_	_NS_	*
	Summer 2019							
1	-	0.006	-	0.007	0.050	0.134	0.432 b	0.608 ab
2	0.002	0.008	-	0.050	0.309	0.170	1.193 a	0.780 ab
3	-	0.030	0.002	0.026	0.084	0.235	0.379 b	0.528 b
4	-	0.000	-	0.010	0.087	0.211	0.374 b	1.194 a
Significance	_NS_	_NS_	_NS_	_NS_	_NS_	_NS_	**	*
Treatments	Fall 2019							
1	0.059	0.016	-	0.018	0.060	0.125	0.191 b	0.303
2	0.023	0.114	-	0.003	0.250	0.176	0.625 a	0.546
3	-	0.000	-	0.002	0.006	0.103	0.258 b	0.413
4	0.005	0.014	-	0.046	0.164	0.184	0.233 b	0.553

^1^ Hamlin citrus trees budded on either Cleopatra (Cleo) or Swingle (Swc) rootstocks. ^2^ Treatments (T): untreated control (1), full Ca dose (2), full Mg dose (3), and half Ca and half Mg doses (4), (full dose = 45 kg ha^−1^) applied as split application during the flush earl spring (Feb.), summer (Jun.), and late summer (Sep.) of each year. ^3^ The average seasonal: spring (Jan.–May), summer (Jun.–Sep.), and fall (Oct.–Dec.) FRLD means (*n* = 8 trees) followed by different lower case letters are significantly different at *p*
< 0.05, based on the Tukey–Kramer honestly significant difference (HSD) test. ^4^ NS, *, **, and *** represent non-significant or significant at *p* ≤ 0.05, < 0.01, and < 0.0001, respectively. ^5^ No data was recorded.

**Table 3 plants-09-00483-t003:** The effect of nitrogen and foliar and/or ground-applied micronutrients (Mn, Zn, and B) on root length density of different root classes of HLB-affected Hamlin citrus trees during 2018–2019 seasons at Immokalee, FL.

Root Diameter (mm)	Spring 2018				Spring 2019			
	<0.5	0.5–1	1–2	>2	<0.5	0.5–1	1–2	>2
Treatments ^1^	Root length density (cm·cm^-2^)							
1	0.012 ^2^	0.024	0.130	0.561 a	0.004	0.005	0.045	0.751
2	0.015	0.036	0.089	0.175 ab	0.023	0.030	0.081	0.958
3	0.027	0.039	0.093	0.301 ab	0.081	0.008	0.161	0.805
4	0.005	0.001	0.029	0.115 b	0.095	0.000	0.070	0.257
Significance ^3^	_NS_	_NS_	_NS_	*	_NS_	_NS_	_NS_	_NS_
Treatments	summer 2018				summer 2019			
1	0.181	0.151	0.207	0.583 a	0.000	0.000	0.010	0.769 a
2	0.024	0.036	0.105	0.398 ab	0.001	0.013	0.078	0.937 a
3	0.023	0.032	0.179	0.754 a	0.032	0.019	0.053	0.674 a
4	0.003	0.026	0.101	0.182 b	0.004	0.000	0.015	0.286 b
Significance	_NS_	_NS_	_NS_	**	_NS_	_NS_	_NS_	**
	fall 2018				fall 2019			
1	0.002	0.014	0.079	0.692	0.000	0.005	0.020	0.263 a
2	0.050	0.007	0.076	0.655	0.001	0.006	0.044	0.315 a
3	0.054	0.021	0.130	0.592	0.000	0.001	0.004	0.275 a
4	0.038	0.001	0.022	0.218	0.009	0.004	0.029	0.114 b
Significance	_NS_	_NS_	_NS_	**	_NS_	_NS_	_NS_	**
Treatments	2018				2019			
	ANOVA							
Factorial effect ^4^	<------------------------------------ Significance ----------------------------------->							
T	_NS_	*	*	***	_NS_	_NS_	**	***
N	_NS_	_NS_	_NS_	_NS_	_NS_	_NS_	_NS_	_NS_
M	_NS_	_NS_	_NS_	**	_NS_	_NS_	_NS_	***
N × M	_NS_	_NS_	_NS_	_NS_	_NS_	_NS_	_NS_	_NS_
M × T	_NS_	_NS_	_NS_	*	_NS_	_NS_	_NS_	_NS_
N × M	_NS_	_NS_	_NS_	_NS_	_NS_	_NS_	_NS_	_NS_
N × M × T	_NS_	_NS_	_NS_	_NS_	_NS_	_NS_	_NS_	_NS_

^1^ Treatments: (1) 0× (untreated control), (2) 1× (foliar only), (3) 2× (1× foliar and 1× ground-applied), and (4) 3× (1× foliar and 2× ground-applied; 1× = 9 kg ha^-1^ of metallic S encapsulated Mn and Zn each and 2.2 kg ha^−1^ of B) applied as split application during the flush earl spring (Feb.), summer (Jun.), and late summer (Sep.) of each year. ^2^ The average seasonal: spring (Jan.–May), summer (Jun.–Sep.), and fall (Oct.–Dec.) FRLD means (*n* = 8 trees) followed by different lower case letters are significantly different at *p*
< 0.05, based on the Tukey–Kramer HSD test. ^3^ NS, *, **, and *** represent non-significant or significant at *p* ≤ 0.05, < 0.01, and < 0.0001, respectively. ^4^ Factorial effects: T = season of the year, N = nitrogen rate, and M = micronutrients, reading was taken every month from Jan. to Dec. of 2018 and 2019.

**Table 4 plants-09-00483-t004:** Root survival (lifespan) analysis using the Kaplan–Meier survival model of citrus trees cv. Hamlin budded on Cleo (A) or Swc (B) rootstocks during the 2018 and 2019 growing seasons.

Event Time				Number of Live Roots at Risk				Number of Roots Died over Time				Root Survival Probability				Standard Error			
**Secondary macronutrients treated Hamlin citrus trees budded on Cleo rootstocks ^1^ (A)**																			
**1**	**2**	**3**	**4**	**1**	**2**	**3**	**4**	**1**	**2**	**3**	**4**	**1**	**2**	**3**	**4**	**1**	**2**	**3**	**4**
30	248	61	189	12	12	12	12	1	1	2	1	0.92	0.92	0.83	0.92	0.08	0.08	0.11	0.08
33	301	153	209	11	11	10	11	2	1	2	1	0.75	0.83	0.67	0.83	0.13	0.11	0.14	0.11
59	307	189	224	9	10	8	10	1	1	2	1	0.67	0.75	0.50	0.75	0.14	0.13	0.14	0.13
61	310	217	238	8	9	6	9	3	1	1	1	0.42	0.67	0.42	0.67	0.14	0.14	0.14	0.14
93	322	245	267	5	8	5	8	3	1	1	1	0.17	0.58	0.33	0.58	0.11	0.14	0.14	0.14
94	325	311	285	2	7	4	7	1	1	1	1	0.08	0.50	0.25	0.50	0.08	0.14	0.13	0.14
120	331	312	312	1	6	3	6	1	1	1	2	0.00	0.42	0.17	0.33	0.00	0.14	0.11	0.14
	334	339	320		5	2	4		1	1	1		0.33	0.08	0.25		0.14	0.08	0.13
	340	344	332		4	1	3		1	1	1		0.25	0.00	0.17		0.13	0.00	0.11
	342		339		3		2				1				0.08				0.08
	349		345		2		1				1				0.00				0.00
**Secondary macronutrients rates of Hamlin citrus trees budded on Swc rootstocks (B)**																			
88	185	95	120	12	12	12	12	1	1	2	1	0.92	0.92	0.83	0.92	0.08	0.08	0.11	0.08
120	215	153	189	11	11	10	11	1	2	1	2	0.83	0.75	0.75	0.75	0.11	0.13	0.13	0.13
127	248	218	218	10	9	9	9	1	1	1	2	0.75	0.67	0.67	0.58	0.13	0.14	0.14	0.14
153	283	248	220	9	8	8	7	1	1	3	1	0.67	0.58	0.42	0.50	0.14	0.14	0.14	0.14
189	300	278	248	8	7	5	6	1	1	1	3	0.58	0.50	0.33	0.25	0.14	0.14	0.14	0.13
245	330	283	252	7	6	4	3	1	1	1	1	0.50	0.42	0.25	0.17	0.14	0.14	0.13	0.11
248	339	312	320	6	5	3	2	1	1	1	1	0.42	0.25	0.17	0.08	0.14	0.13	0.11	0.08
263	345	330	341	5	3	2	1	1	1	1	1	0.33	0.17	0.08	0.00	0.14	0.11	0.08	0.00
275	352	344		4	2	1		1	1	1		0.25	0.08	0.00		0.13	0.08	0.00	
278	358			3	1			1	1			0.17	0.00			0.11	0.00		
283	378			2				1				0.08				0.08			
312				1				1				0.00				0.00			

^1^ Treatments: untreated control (1), full Ca dose (2), full Mg dose (3), and half Ca and half Mg doses (4), (full dose = 45 kg ha^−1^).

**Table 5 plants-09-00483-t005:** The FRLD survival (lifespan) analysis using the Kaplan–Meier survival for the survival curves of citrus trees cv. Hamlin budded on Cleo or Swc rootstocks and Valencia citrus trees budded on Swc rootstocks.

Scion	Hamlin Citrus								Valencia Citrus			
Rootstocks	Cleo				Swc				Swc			
	Secondary macronutrients ^1^						Micronutrients ^2^					
Treatments	1	2	3	4	1	2	3	4	1	2	3	4
Mean ^3^	69 ± 8	321 ± 8	214 ± 29	281 ± 27	215 ± 22	334 ± 29	244 ± 27	246 ± 25	269 ± 14	298 ± 17	315 ± 14	187 ± 25
	<---------------------------------------------------------------------------------- Percentiles ---------------------------------------------------------------------------------- >											
25	93 ± 14	340 ± 6	311 ± 50	320 ± 15	275 ± 11	408 ± 16	283 ± 26	248 ± 12	312 ±17	335 ± 18	335 ± 119	249 ± 10
50 (Median) ^4^	61 ± 1	325 ± 8	189 ± 37	285 ± 30	245 ± 51	339 ± 82	248 ± 17	220 ± 13	254 ± 20	294 ± 27	316 ± 22	191 ± 52
75	33 ± 14	307 ± 7	153 ± 75	224 ± 21	127 ± 25	215 ± 32	153 ± 92	189 ± 37	225 ± 16	270 ± 14	285 ± 46	111 ± 22
95% C.I.	53–85	305–337	158–271	253–360	172–258	277–390	191–298	198–294	241–298	265–333	288–343	138–236
Significance ^5^	<0.001				0.095				<0.001			

^1^ Secondary macronutrients: untreated Control (1), full Ca dose (2), full Mg dose (3), and half Ca and half Mg doses (4), (full dose = 45 kg ha^−1^). ^2^ Micronutrients: (1) 0× (untreated control), (2) 1× (foliar only), (3) 2× (1× foliar and 1× ground-applied), and (4) 3× (1× foliar and 2× ground-applied), (1× = 9 kg ha-1 of metallic S encapsulated Mn and Zn each and 2.2 kg ha-1 of B). ^3^ The mean of twelve roots±SEM. ^4^ The median survival time [27] (lifespan) is the survival time at which the root survival probability ≤ 50% ± SEM. ^5^ The Gehan–Breslow statistic for the survival curves showed a statistically significant difference between survival curves (*p* < 0.001).

**Table 6 plants-09-00483-t006:** The FRLD survival analysis using the Kaplan–Meier survival model of citrus trees cv. Valencia budded on Swc rootstocks during 2018 and 2019 growing seasons.

Event Time				Number of Live Roots at Risk				Number of Roots Died over Time				Root Survival Probability				Standard Error			
Secondary macronutrients treated Hamlin citrus trees budded on Swc rootstocks ^1^																			
**1**	**2**	**3**	**4**	**1**	**2**	**3**	**4**	**1**	**2**	**3**	**4**	**1**	**2**	**3**	**4**	**1**	**2**	**3**	**4**
181	156	218	30	12	12	12	12	1	1	1	1	0.92	0.92	0.92	0.92	0.08	0.08	0.08	0.08
218	249	249	94	11	11	11	11	1	1	1	1	0.83	0.83	0.83	0.83	0.11	0.11	0.11	0.11
225	270	285	111	10	10	10	10	1	1	1	1	0.75	0.75	0.75	0.75	0.13	0.13	0.13	0.13
249	276	310	123	9	9	9	9	2	2	1	1	0.58	0.67	0.67	0.67	0.14	0.14	0.14	0.14
254	294	314	156	7	7	8	8	1	1	1	1	0.50	0.58	0.58	0.58	0.14	0.14	0.14	0.14
272	307	316	191	6	6	7	7	1	1	2	1	0.42	0.42	0.50	0.50	0.14	0.13	0.14	0.14
285	335	333	216	5	5	5	6	1	2	1	1	0.25	0.33	0.42	0.42	0.13	0.11	0.11	0.14
312	342	335	233	4	3	4	5	2	1	1	1	0.17	0.25	0.33	0.33	0.11	0.00	0.08	0.14
335	373	370	249	2	2	3	4	1		2	2	0.00		0.17	0.17	0.08		0.00	0.11
345		373	252	1		1	2	1			1	0.92			0.08	0.00			0.08
			340				1				1				0.00				0.00

^1^ Treatments: (1) 0× (untreated control), (2) 1× (foliar only), (3) 2× (1× foliar and 1× ground-applied), and (4) 3× (1× foliar and 2× ground-applied), (1× = 9 kg ha^−1^ of metallic S encapsulated Mn and Zn each and 2.2 kg ha^−1^ of B).

**Table 7 plants-09-00483-t007:** Effects of ground-applied plant nutrition on the root-zone soil pH of HLB-affected Hamlin citrus trees in Immokalee, FL during the 2017–2019.

	2017				2018				2019			
	Spring		Summer		Spring		Summer		Spring	Summer	Spring	Summer
	Soil depth (0–15 cm)											
Treatments ^1^	Cleo	Swc	Cleo	Swc	Cleo	Swc	Cleo	Swc	Cleo	Swc	Cleo	Swc
1	7.62 ab^2^	7.47 b	7.05 a	6.91 a	6.68 a	6.57 a	6.08 a	6.16 a	6.37	6.22	6.38 a	6.13 ab
2	7.67 ab	7.56 ab	6.97 ab	6.83 a	6.45 ab	6.45 a	6.10 ab	6.04 a	6.53	6.15	6.35 a	6.17 ab
3	7.64 b	7.72 a	7.05 ab	6.94 a	6.59 ab	6.57 a	5.98 b	6.11 a	6.40	6.33	6.14 a	6.02 b
4	7.72 a	7.63 ab	6.84 b	6.86 a	6.54 b	6.43 a	6.13 a	6.10 a	6.51	6.26	6.19 a	6.17 a
Significance ^3^	*	**	**	_NS_	**	_NS_	*	_NS_	_NS_	_NS_	_NS_	*
	Soil depth (15–30 cm)											
1	7.70	7.51	7.18	7.06	6.76	6.73	6.52	6.63	6.68	6.57	6.25	6.06
2	7.67	7.64	6.98	7.10	6.55	6.61	6.34	6.49	6.57	6.54	6.25	5.83
3	7.56	7.74	7.00	7.14	6.66	6.65	6.21	6.38	6.44	6.60	6.08	5.71
4	7.77	7.66	6.97	7.13	6.61	6.62	6.28	6.55	6.63	6.64	5.99	6.16
Significance	_NS_	_NS_	_NS_	_NS_	_NS_	_NS_	_NS_	_NS_	_NS_	_NS_	_NS_	_NS_
	Soil depth (30–45 cm)											
1	7.67	7.56	7.21	7.15	6.91	6.84	6.70	6.63	6.65	6.57	6.24	6.01
2	7.02	7.59	7.05	7.17	6.17	6.75	5.94	6.61	6.07	6.41	6.21	6.02
3	7.58	7.78	6.96	7.16	6.86	6.83	6.39	6.45	6.54	6.62	6.08	5.81
4	7.82	7.64	7.04	7.24	6.72	6.81	6.39	6.63	6.82	6.56	6.10	6.33
Significance	_NS_	_NS_	_NS_	_NS_	_NS_	_NS_	_NS_	_NS_	_NS_	_NS_	_NS_	_NS_
	ANOVA											
Effect ^4^	<-----------------------------------------------------Significance ^3^ ------------------------------------------------------>											
D	_NS_		***		***		***		***		*	
R	_NS_		***		***		**		*		**	
N	***		_NS_		_NS_		**		_NS_		_NS_	
S	*		_NS_		*		**		_NS_		_NS_	
D × R	_NS_		*		*		_NS_		_NS_		_NS_	
D × N	_NS_		_NS_		_NS_		_NS_		_NS_		_NS_	
D × S	_NS_		_NS_		_NS_		_NS_		_NS_		_NS_	
R × S	**		_NS_		_NS_		_NS_		_NS_		*	
R × N	_NS_		_NS_		_NS_		_NS_		_NS_		_NS_	
N × S	**		*		*		***		*		_NS_	
D × R × N	_NS_		_NS_		_NS_		_NS_		_NS_		_NS_	
D × N × S	_NS_		_NS_		_NS_		_NS_		_NS_		_NS_	
D × R × S	_NS_		_NS_		_NS_		_NS_		_NS_		_NS_	
N × R × S	_NS_		***		**		***		_NS_		_NS_	
D × R × N × S	_NS_		_NS_		_NS_		_NS_		_NS_		_NS_	

^1^ Treatments: untreated control (1), full Ca dose (2), full Mg dose (3), and half Ca and half Mg doses (4), (full dose = 45 kg ha^−1^). ^2^ Means on vertical column followed by different letters are significantly different at *p* ≤ 0.05, based on the Tukey–Kramer HSD test (*n* = 36). ^3^ NS, *, **, and *** represent non-significant or significant at *p* ≤ 0.05, < 0.01, and < 0.0001, respectively. ^4^ Factorial effects: D = depths of the soil, R = rootstocks, N = nitrogen rate, and S = secondary micronutrients.

**Table 8 plants-09-00483-t008:** Effects of ground-applied plant nutrition on the root-zone soil pH of HLB-affected Valencia citrus trees in Immokalee, FL during the 2017–2019.

	2017		2018		2019	
	Spring	Summer	Spring	Summer	Spring	Summer
Treatments ^1^	Soil depth (0–15 cm)					
1	7.23 a^2^	7.18 a	6.27 a	6.02 a	5.98 a	5.91 ab
2	7.12 a	7.09 a	6.47 a	5.86 a	5.64 ab	6.35 a
3	6.82 ab	6.70 ab	4.78 b	5.02 b	5.05 bc	5.79 ab
4	6.38 b	6.22 b	4.57 c	3.78 c	4.44 c	5.36 b
Significance ^3^	**	***	***	***	***	**
	Soil depth (15–30 cm)					
1	7.45 a	7.36 a	6.58 a	6.31 a	6.27 ab	5.99 ab
2	7.30 ab	7.33 a	6.63 a	6.16 a	6.34 a	6.38 a
3	7.18 ab	7.00 ab	5.67 b	5.44 b	5.64 bc	5.49 bc
4	6.97 b	6.40 b	5.11 c	4.17 c	4.92 c	4.99 c
significance	*	***	***	***	***	***
	Soil depth (30–45 cm)					
1	7.38 a	7.47 a	6.77 a	6.42 a	6.50 a	6.33 ab
2	7.27 ab	7.45 a	6.80 a	6.08 a	6.63 a	6.59 a
3	7.03 ab	7.34 a	6.07 b	5.82 a	6.25 ab	5.70 bc
4	6.74 b	6.94 a	5.01 b	4.51 b	5.74 b	5.20 c
significance	*	ns	***	***	**	***
	ANOVA					
Effect ^4^	<----------------------Significance level------------------------>					
D	***	***	***	***	***	_NS_
N	_NS_	_NS_	_NS_	_NS_	_NS_	_NS_
M	***	***	***	***	***	***
D × N	_NS_	_NS_	_NS_	_NS_	_NS_	_NS_
D × M	*	_NS_	*	_NS_	_NS_	_NS_
N × M	_NS_	_NS_	_NS_	_NS_	_NS_	_NS_
D × N × M	_NS_	_NS_	_NS_	_NS_	_NS_	_NS_

^1^ Treatments: (1) 0× (untreated control), (2) 1× (foliar only), (3) 2× (1× foliar and 1× ground-applied), and (4) 3× (1× foliar and 2× ground-applied), (1× = 9 kg ha^−1^ of S encapsulated metallic Mn and Zn each and 2.2 kg ha^−1^ of B). ^2^ Means on vertical column followed by different letters are significantly different at *p* < 0.05, based on the Tukey-Kramer HSD test (*n* = 36). ^3^ Significance: NS, *, **, and *** non-significant or significant at *p* ≤ 0.05, < 0.01, and < 0.0001, respectively. ^4^ Factorial effects: D = depths of the soil, N = nitrogen rate, and M = micronutrients.

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
