# Peer review of "Effect of Essential Nutrients on Roots Growth and Lifespan of Huanglongbing Affected Citrus Trees"

_plants, 2020, doi:10.3390/plants9040483_

Round 1
Reviewer 1 Report
Please, see attached file.

Author Response
Response to Reviewer 1 Comments
Overall this is an interesting paper but there are some major issues that need to be fixed before considering publication.
Major issues
First of all, a re-thinking of the root diameter classification have to be considered. It is becoming common to divide the “feeder” part of woody-plant root systems in fibrous and pioneer roots, the first one used to absorption and the second one for exploring soil, transport water/nutrients and to generate fibrous roots. Fibrous roots are more short-lived and of smaller diameters than pioneer roots. D. M. Eissenstat and D. S. Achor in 1999 published an interesting study about roots of citrus rootstocks: “Anatomical characteristics of roots of citrus rootstocks that vary in specific root length”. In their work, “(…) Roots were divided into three categories: first-order fibrous roots (terminal roots with growing root tips), second-order fibrous roots (portions of the root that bear first- order roots) and pioneer roots (large, straight, generally unbranched roots with prominent tip and large diameter “. Moreover, I found more interesting that “(…) A frequency distribution of diameter of field first-order roots of TO was distinctly bimodal, with peaks at 400 and 500 μm. Frequency distributions of other rootstocks exhibited first-order root diameters primarily at 500 μm (550 μm for SO) with secondary peaks between 650 and 900 μm. In addition, the diameter of pioneering roots range between 0,854 to 1,292 mm. In another study, “Huanglongbing-induced Anatomical Changes in Citrus Fibrous Root Orders” (2018) of N. Kumar and F. Kiran, four root orders were defined on the base of topological analysis (see also pictures herein): “The diameter of roots within various root orders declined from the fourth order to the first order [fourth: 1.2 (± 0.01) mm; third: 0.6 (± 0.02) mm; second: 0.5 (± 0.01) mm; first: 0.5 (± 0.1) mm]”. So, based on the previous notes, the division made by authors should be improved.
Response 1: As per your comments, the results on Table 2 had been reconsidered and categorized as the customary root classification. I tried to check the accuracy of the minirhizotron measurements; the accuracy level is ≈99%. The figures indicated in this specific table is not the actual root lengths or diameter rather they are the density of root (cm) per unit cm2. This figures are relatively similar to the field experiment done by Kadyampakeni et al., 2014. “Effect of Irrigation Pattern and Timing on Root Density of Young Citrus Trees…… “
Secondly, based on my experience about woody-plant fine root systems, I found weird your data about RLD, and it seemed to me stranger after the quick research I done about citrus fine roots. The strangest thing is the very low amount of roots under 1.0 mm diameter (where it is expected to find the highest amount in terms of length). And secondly, the big amount of roots with diameter > of 3 mm. Seems like your data are shifted, like there was a wrong estimation of root diameters from your acquisitions. And it is not possible to assert that it is an effect of Huanglongbing-affection, because in the study 1) where not analysed healthy trees to make comparisons and 2) also in HLB tree, as shown in Fig.2 of N. Kumar and F. Kiran (2018), the finest roots should be more than bigger ones. I suggest to authors to check if calibrations of hardware and software are correct, maybe making some tests measuring known-sized object (that could be put near the walls of the tube of the minirhizotron).
Response 2: I agree on your point that the fibrous root number should be more than the structural roots (>2.0mm) in woody-plant or perennial trees. This is the epicenter of the study on which hypothesis were built on. On young citrus trees we don’t have such problems. On mature, field experiments, and grower groves the decline in fine roots has been prevalent a decline of ≈70 -80% of the fine roots. Moreover, it is almost impossible to get HLB-free citrus tree in the Florida citrus industry. The trees in the current study are 100% infected with Candidatus Liberibacter asiaticus. Laboratory results performed with quantitative real-time polymerase chain reaction (qPCR) indicated that both sets of trees were positive for the HLB associated bacterium, Ca. Las with Ct value of 25.3 ± 0.25 and 25.9 ± 0.19 of the Hamlin and Valencia trees, respectively. Therefore, our study focused if we treat trees with essential nutrients, as most fine roots are affected by the disease, will reverse the decline in fibrous roots density; thus the industry maintain producing quality fruits. Our comparison was between the essential nutrient treated trees with that of the control untreated trees.
Minor issues
Introduction and material and methods
Lines 64-66. In the first sentence it is states that ground-applied fertilizer reduces leaching while the following sentence lists leaching as a problem of ground-applied fertilizer.
Re-wrote as:
Response 3: Ground-applied fertilizers are subject to unfavorable soil processes such as precipitation as forms that are chemically unavailable to plants, leaching, and runoff [3,18]. Therefore, split ground-applied and foliar-applied fertilizer improves nutrient use efficiency and reduces both nutrient leaching and application cost [17].
Reference n. 16 has to be removed because it is not related to the sentence.
Response 4: Removed.
Lines 82-84. It’s missing a bracket “(“
Response 5: Edited
Line 116. How many minirhizotron tubes were installed in total for each treatment? Please add.
Response 6: Eight minirhizotron tubes were installed per treatment in March 2017… total of 32 minirhizotron per experiment.
Line 118. Avoiding root damage when digging is utopian. I’ll rephrase like “(…) auger to reduce soil disturbance and roots damage”.
Response 7: Edited to: … auger to reduce soil disturbance and roots damage.
Figure 1. In the top view is not immediately clear what are 12 m and 3 m (I suppose spacing between rows and plants, respectively). Should be stated somewhere, on the drawing or in the caption.
Response 8: Edited to … with 12m and 3m spacing between rows and trees, respectively.
Line 147. Seems that the sentence “For F-tests with a statistical difference at (P<0.05)” is missing of something. Please rephrase.
Response 9: Rephrased to: For F-tests with a statistical difference at (P<0.05) was used and the Tukey-Kramer Honestly Significant Difference (HSD) grouping range test was used to compare the means.
Line 148. Why is it chosen this diameter class for root survival analysis?
Response 10: Roots on this category significantly responded to the treatment applied. Therefore, this root category was picked to show the coherence of the treatment. Root survival (lifespan) analysis was in accordance with this root category.
Results and discussion.
I didn’t deepened too much the revision of these sections because of the major issues of this work.
Response 11: The discussion was reviewed in line to the major issues addressed.
Line 164. It is not clear how percentages were calculated (I tried based on data in tables but sometimes I obtained different results). Maybe should be better to specified it.
Response 12: Previously, I made the %-age per the annual basis. This will be difficult for the reader to follow. Now, I calculated the %-age per the specific season of the root category stated in the sentence in accordance with the above justification (Line 148).
% = (the root-order of the season FRLD /the total root order (first – fourth) FRLD × 100
Line 168-169. This sentence refers to data completely different from the ones treated in the work, percentages of roots in different soil profiles compared to percentage of roots referring to different roots diameter classes. This sentence have to be removed and at this point also the reference n. 16 in the bibliography list.
Response 13: Removed.
Line 186. Only at this point the term “structural root” appear and in line 194 is clarified. But based on major issues, structural roots should not appear anymore.
Response 14: Reviewed as per the comment.
Lines 232-234. I suggest some improvements in the discussion. In general, a double peak is related to the phenology of the plant and its behavior related to environmental factor (e.g. dry summers). A first peak could be related to water and nutrient acquisition for the seasonal growth and the second peak for the nutrient acquisition to increase plant reserves before winter. Montagnoli et al 2019 (and reference herein) “Seasonality of fine root dynamics and activity of root and shoot vascular cambium in a Quercus ilex L. forest (Italy)” are good example of bimodal patterns in Mediterranean area.
Response 15: Reviewed: I agree and cited as suggested. Florida has similar to the Mediterranean climate: it has a bimodal rainfall patter (April – May) with a brief dry season (August – September) another peak rainfall seasons.
Line 235. Please change “phenology” instead of “phonology”.
Response 16: Edited: to … attributed to the phenology of the trees.
Line 262. “In summer 2019, the foliar (1X) and foliar (1X) treatment…” It is missing something.
Response 17: Reviewed: In summer 2019, the [foliar (1X) and soil (1X)]….
Conclusion
Line 380. How can authors assert this sentence? In the experimental design are missing healthy tree without HLB, so, there isn’t this kind of comparison.
Response 18: Similar justification as above. The experiments are set as treated with essential nutrients with those control untreated trees. It is less likely to get a health mature tree in the Florida citrus groves.

Reviewer 2 Report
This manuscript aims at finding a positive way to prevent plant dieback in infected trees with Candidatus liberibacter. Particularly, the authors attempted at determining the impact of of soil and foliar-applied essential nutrients on the structural and root length density over the seasons, monthly RLD pattern, RLD survival probability, and soil acidity of citrus trees infected by the afore mentioned bacteria.
The idea of amending the affected plants with the needed nutrients as a way to maintain their production is very interesting a possible a good practice before a treatment against the bacteria is found. Results from this study could add lots of new information to the study of poorly known infective plant pathogens. There several comments below that will help improve the manuscript.
The title could be shortened as in its current form is very long.
The abstract can also be reduced and in turn, present conclusions on how to treat the different varieties of citrus to reduce the studied bacterial infection (Huanglongbing).
In the introduction, together with the presented objectives, it is needed to present the hypotheses that are to be tested. This is an experimental study, for what scientific hypotheses are needed, and based on those hypotheses, there will be proper conclusions.
In the Material and Methods section, where the treatments are explain it is necessary to justify the two different amounts of N applied to the trees and similarly, explain the reasons why the concentrations of micronutrients were those and no others.
After having read the introduction one gets the feeling that the study would look for the root development of Huanglongbing infected oranges trees under the proposed fertilization treatments. However, in the experimental design it is not said that the treated treed were or weren’t infected. If the trees were not infected, what is the link between Huanglongbing infection and the actual study? This needs revision otherwise the reader would be disappointed not to find such a relationship. What is more, it would be desired a line indicating the level of infection in the trees, should this be the case. Results would vary depending on the level of infection suffered by the treated trees.
Where the soil samples collected only once?
Tables 1, 2 and 6 are pretty complicated to follow and to understand. Please, simplify them to assist the reader to easily understand the message you want to convey.
In general, the sheer amount of data presented in the results section is difficult to follow. I would suggest that the authors split this section into two: Results one and Discussion another. This way, the discussion of all the results will be clearer and better presented. In the current form, part of the discussion is buried in the text, and some section, like 3.2 and 3.4 are not discussed. Thus, the impressive work behind this manuscript will be properly presented and made available to the scientific community.
On lines 348-9 it is said ‘During the spring season of 2017, the soil pH 348 varied across the three soil depths in reaction to the micronutrient rates’, however there is no proof that the pH variation is only due to the micronutrients applied. This needs to be rewritten.
The conclusion, in is current form is a partial discussion. In the conclusion section the scientific hypothesis would be accepted or rejected, making of this a clear and concise section. Information on lines 380-2 is results. The same happens to the sentence on lines 382-5. This is followed by another two sentences till line 391, also containing a summary of the results previously presented and poorly discussed. Unfortunately, the second paragraph of the Conclusions section bares the same kind of information, not concluding regards anything.
It would be fantastic that the authors could come up with a proposal on how to treat the citrus trees based on the use of directed fertilization as this will be a clear contribution to both, plant science and agriculture.
Author Response
Response to Reviewer 2 Comments
Comments and Suggestions for Authors
This manuscript aims at finding a positive way to prevent plant dieback in infected trees with Candidatus liberibacter. Particularly, the authors attempted at determining the impact of soil and foliar-applied essential nutrients on the structural and root length density over the seasons, monthly RLD pattern, RLD survival probability, and soil acidity of citrus trees infected by the afore mentioned bacteria.
The idea of amending the affected plants with the needed nutrients as a way to maintain their production is very interesting a possible a good practice before a treatment against the bacteria is found. Results from this study could add lots of new information to the study of poorly known infective plant pathogens. There several comments below that will help improve the manuscript.
The title could be shortened as in its current form is very long.
Response 1: Title reduced from 24 to 14 words as: “Effect of essential nutrients on roots growth and lifespan of Huanglongbing affected citrus trees”
The abstract can also be reduced and in turn, present conclusions on how to treat the different varieties of citrus to reduce the studied bacterial infection (Huanglongbing).
In the introduction, together with the presented objectives, it is needed to present the hypotheses that are to be tested. This is an experimental study, for what scientific hypotheses are needed, and based on those hypotheses, there will be proper conclusions.
Response 2: The hypotheses was included in the introduction section.
….Hence, the following hypotheses were tested: 1) foliar and or ground-applied fertilizers improve the root growth of HLB-affected citrus trees in response to essential nutrients, and 2) essential nutrients also boost seasonal FRLD lifespan with emphasis on root-zone soil pH.
In the Material and Methods section, where the treatments are explain it is necessary to justify the two different amounts of N applied to the trees and similarly, explain the reasons why the concentrations of micronutrients were those and no others.
Response 3: Our basis of all fertilizer recommendations is the University of Florida, institute of food and agricultural science (UF/IFAS). The source of information is described and cited in the materials and methods section.
After having read the introduction one gets the feeling that the study would look for the root development of Huanglongbing infected oranges trees under the proposed fertilization treatments. However, in the experimental design it is not said that the treated treed were or weren’t infected. If the trees were not infected, what is the link between Huanglongbing infection and the actual study? This needs revision otherwise the reader would be disappointed not to find such a relationship. What is more, it would be desired a line indicating the level of infection in the trees, should this be the case. Results would vary depending on the level of infection suffered by the treated trees.
Response 4: The trees in the current study are 100% infected with Candidatus Liberibacter asiaticus. Laboratory results performed with quantitative real-time polymerase chain reaction (qPCR) indicated that both sets of trees were positive for the HLB associated bacterium, Ca. Las with Ct value of 25.3 ± 0.25 and 25.9 ± 0.19 of the Hamlin and Valencia trees, respectively. Therefore, our study focused if we treat trees with essential nutrients, as most fine roots are affected by the disease, will reverse the decline in fibrous roots density; thus the industry maintain producing quality fruits. Our comparison was between the essential nutrient treated trees with that of the control untreated trees.
Where the soil samples collected only once?
Response 5: The soil samples were collected at three different location of three separate depths under the tree canopy ≈0.5–1.0m distance from the tree trunk.
Tables 1, 2 and 6 are pretty complicated to follow and to understand. Please, simplify them to assist the reader to easily understand the message you want to convey.
In general, the sheer amount of data presented in the results section is difficult to follow. I would suggest that the authors split this section into two: Results one and Discussion another. This way, the discussion of all the results will be clearer and better presented. In the current form, part of the discussion is buried in the text, and some section, like 3.2 and 3.4 are not discussed. Thus, the impressive work behind this manuscript will be properly presented and made available to the scientific community.
Response 6: I agree that Table 1 look a beat lengthy and difficult for a reader to follow. Therefore, the root-orders and their respective ANOVA table were separated and set as Table 1 and 2.
As per your suggestion, the results and discussion section are separated in two sections, i.e., result and discussion was dealt separately.
On lines 348-9 it is said ‘During the spring season of 2017, the soil pH 348 varied across the three soil depths in reaction to the micronutrient rates’, however there is no proof that the pH variation is only due to the micronutrients applied. This needs to be rewritten.
The conclusion, in its current form is a partial discussion. In the conclusion section the scientific hypothesis would be accepted or rejected, making of this a clear and concise section. Information on lines 380-2 is results. The same happens to the sentence on lines 382-5. This is followed by another two sentences till line 391, also containing a summary of the results previously presented and poorly discussed. Unfortunately, the second paragraph of the Conclusions section bares the same kind of information, not concluding regards anything.
Response 7: The soil pH varied across the three soil depths in reaction to the S encapsulated micronutrient rates, not because of the micronutrients. I miss to include the built-in sulfur to enhance micronutrient uptake.
The conclusion part has been reconsidered in accordance with the hypotheses set at the beginning of the study.
It would be fantastic that the authors could come up with a proposal on how to treat the citrus trees based on the use of directed fertilization as this will be a clear contribution to both, plant science and agriculture.
Submission Date
26 February 2020

Round 2
Reviewer 1 Report
You'll find my comments directly on the attached .pdf
I appreciate the efforts made to correct the manuscript based on my revision, but now there is a main issue about the description of the roots subdivision. It is not right to use the term "root order" because it is not based on a a-priori topological study. It is convenient to use the term "root diameter class". After these changes, the manuscript will sound right.

Author Response
Response to Reviewer 1 Comments round II
Response 1:
Abstract:
Line 14, 17 and 18 the terms used here have been changed to the traditional fine (< 2 mm) and coarse (> 2 mm) root classification as per the recommendation.
Response 2:
Introduction:
The hypotheses on the previous version were in Line 47 – 49; now moved down to line 56 – 59 (highlighted in yellow).
Response 3:
Materials and Methods:
The sentence in line 83 was removed and edited as “The trees in the current study were …. in line 76 – 79.
The sentence in line 146 – 147, now has been re-edited as “The monthly FRLD data were grouped based on traditional root classification: fine roots (d < 2 mm) and coarse root (d > 2 mm)…. (Line 129 – 131).
Response 4:
Results:
Words and phrases in line: 161-164, 169, 171 – 174, are re-edited as per the tradition root classification. See highlighted phrases in line: 153, 154, 155, 158, 159, 161, 163, 183, 185, 193, 204, 223, and 282, and 283.
Response 5:
Discussions: Words and phrases in line: 360, 361, 364, 365, 369, 372, 375, 419, 429, 432, and 434 are reedited as per the tradition root classification.
See highlighted the reviewed phrases in line: 358, 360, 362, 363, 368, 371, and 373.

Reviewer 2 Report
Thank you very much for the effort to amend the manuscript following my suggestions. I do acknowledge that most of them have been taken into account. The manuscript is almost there, but still few questions need to be addressed.
The abstract is exactly the same as it was, i.e. very long and a reduction is desired. Thanks
Thanks for the hypotheses in the introductions. There is a common mistake when producing them which is to present predictions rather than actual hypotheses. Could you please transform them into real hypotheses instead of predictions please? Reading the information available in this link could aid the author in this (non easy) task.
https://www.trentu.ca/academicskills/how-guides/how-succeed-math-and-science/writing-science/understanding-hypotheses-and-predictions
The information on lines 85and 86 is still unclear. Please, write there that ‘The trees in the current study are 100% infected with Candidatus Liberibacter asiaticus’. Thank you.
Other than that, the manuscript contains useful information that deserves publication.
Author Response
Response to Reviewer 2 Comments round II
Comments and Suggestions for Authors
Thank you very much for the effort to amend the manuscript following my suggestions. I do acknowledge that most of them have been taken into account. The manuscript is almost there, but still few questions need to be addressed.
The abstract is exactly the same as it was, i.e. very long and a reduction is desired. Thanks
Response 1: The abstract has been reduced from 360 words to 231 words. (Line 9–23)
Thanks for the hypotheses in the introductions. There is a common mistake when producing them which is to present predictions rather than actual hypotheses. Could you please transform them into real hypotheses instead of predictions please? Reading the information available in this link could aid the author in this (non easy) task.
https://www.trentu.ca/academicskills/how-guides/how-succeed-math-and-science/writing-science/understanding-hypotheses-and-predictions
Response 2: The hypotheses was reformulated as per the recommendation. (Line 55–59).
The information on lines 85and 86 is still unclear. Please, write there that ‘The trees in the current study are 100% infected with Candidatus Liberibacter asiaticus’. Thank you.
Response 3: ‘The trees in the current study…was taken as it and inserted in line 77 – 79.
Other than that, the manuscript contains useful information that deserves publication.
